# The relationship between interpersonal distance preference and estimation accuracy in autism

**Nur Givon-Benjio**[1]*, **Tom Marx**[1], **Marissa Hartston**[2], **Idan M. Aderka**[1], **Bat-Sheva Hadad**[2], **Hadas Okon-Singer**[1,3,4]

**1** School of Psychological Sciences, University of Haifa, Haifa, Israel, **2** Department of Special Education, University of Haifa, Haifa, Israel, **3** The Integrated Brain and Behavior Center (IBBRC), University of Haifa, Haifa, Israel, **4** The Data Sciences Research Center (DSRC), University of Haifa, Haifa, Israel

* nur.benjio@gmail.com

**Data Availability Statement:** Data and materials are openly available on the project's Open Science Framework page https://osf.io/n7yfr/.

## Abstract

People naturally seek an interpersonal distance that feels comfortable, striking a balance between not being too close or too far from others until reaching a state of equilibrium. Previous studies on interpersonal distance preferences among autistic individuals have yielded inconsistent results. Some show a preference for greater distance, while others indicate a preference for shorter distances, or reveal higher variance in preferences among autistic individuals. In a related vein, previous studies have also investigated the way autistics accurately judge distance, and these studies have received inconsistent results, with some showing superior spatial abilities and others indicating biases in distance estimations. However, the link between distance estimation and preference has never been examined. To address this gap, our study measured interpersonal distance preferences and estimations and tested the correlation between the two factors. The results indicate greater variance among autistic people in both the preference of distance and the ability to estimate distance accurately, suggesting that inconsistencies in previous studies originate from greater individual differences among autistics. Furthermore, only among autistic individuals were interpersonal distance preference and estimation bias associated in a manner that violated equilibrium. Underestimation bias (judging others as closer than they are) was linked to a preference for closer proximity, while overestimation bias (judging others as further away) was associated with a preference for maintaining a greater distance. This connection suggests that biases in the estimation of interpersonal distance contribute to extreme preferences (being too close or too far away). Taken together, the findings suggest that biases in the estimation of interpersonal distance are associated with socially inappropriate distance preferences among autistics.

**Funding:** This work was supported by a grant from the Ministry of Science & Technology, Israel, the Ministry of Europe and Foreign Affairs (MEAE) and the Ministry of Higher Education, Research and Innovation (MESRI) of France [grant number 100008363 awarded to H.O-S.].

**Competing interests:** The authors have declared that no competing interests exist.

# 1. Introduction

## 1.1 Preference for interpersonal distance in autism

Interpersonal distance, defined as the physical space between people [1], constitutes a significant aspect of social interactions. Individuals modulate interpersonal distance through a dynamic interplay of approach and avoidance until achieving a state of equilibrium, where they feel comfortable with their physical proximity to social partners [2–4]. Previous studies have underscored the importance of maintaining an appropriate interpersonal distance for effective communication [5, 6] and safeguarding individuals from potential harm [7]. In the context of autism, abnormalities in interpersonal distance preferences are directly linked to the severity of social deficits [8]. However, existing literature presents a mixed picture regarding the specific direction of these abnormalities. While some studies suggest that autism is associated with a preference for maintaining a larger interpersonal distance [8–11], others propose that autism is linked to a preference for closer physical proximity [12–15], or that it is associated with a larger variance in preferred distance compared to non-autistic individuals [16–18]. Considering that challenges in interpersonal distance regulation not only play a pivotal role in everyday behavior but also contribute to the severity of social deficits among individuals with autism [19], shedding light on the processes underlying distance regulation in autism becomes a matter of paramount importance.

Recent theoretical work [20] lays out how internal factors (such as sensory hypersensitivity and cognitive processing differences) and external factors (such as social context and cultural norms) can interact to shape the interpersonal distance preferences of individuals with autism. For example, the preferred distance might be determined by autistics' hypersensitivity to sensory modalities, making the smell, sound, and appearance of the social partner much more salient. Farkas et al. [20] also suggest that interpersonal distance regulation plays a role in determining the social and cognitive performance of individuals with autism, further highlighting the importance of studying interpersonal distance in this population.

## 1.2 Accuracy of distance estimations in autism

Inconsistencies also exist in the literature regarding the ability of autistics to accurately judge distances. Some studies suggest superior estimation abilities among autistics when estimating the distance from neutral objects. This superiority is evident in their capacity to accurately replicate distances [21], provide accurate verbal estimates of distances [22], and even demonstrate exceptional abilities in a standardized optometric test [23] (although controversial [24, 25]). On the contrary, accumulating evidence points to biases in the visual perception of distance among autistics. For instance, Linkenauger et al. [26] found that autistics tend to overestimate the distance required to reach and grab an object placed in front of them (peripersonal distance; [26], Task 1, supplementary materials). Another experiment indicated that when asked to replicate a specific distance, such as drawing a line of a predetermined length, autistic individuals drew a shorter line than their non-autistic counterparts [27], suggesting a potential underestimation of the line's length. To the best of our knowledge, only one study has examined estimations of interpersonal distance, finding that, following cooperative tool training, autistics tended to overestimate the distance from a confederate [28, although note that the differences in estimation accuracy at baseline are not reported, only the pre and post-training difference]. Therefore, a significant inconsistency exists in the literature regarding the ability of autistics to accurately estimate distance, with most evidence focused on estimating distance from objects and limited evidence on interpersonal distance. Interestingly, the link between the ability to judge distance and the actual distance preference has never been examined.

Given previous studies that linked biases in interpersonal distance estimations with abnormal preferences for distance [29, 30], it is crucial to examine the association between distance estimation and preference to gain insight into the social difficulties faced by autistics.

### 1.3 The current study

In order to address this gap, we aimed to investigate interpersonal distance preference, interpersonal distance estimation, and the relationships between these factors among both autistic and non-autistic participants. To do so, we designed a novel computerized task utilizing high-quality videos that simulate social interactions. Given the inconsistencies in the literature, our study took an exploratory approach.

## 2. Method

### 2.1 Participants

The research cohort included 80 participants: 39 with a confirmed diagnosis of Autism Spectrum Disorder (30 men, 5 women, 4 none/other; $M_{age}$ = 26.85, $SD_{age}$ = 6.18, Range = 18–50; 34 right-handed, 4 left-handed, 1 ambidextrous; all high functioning) and 41 non-autistic participants (12 men, 29 women; $M_{age}$ = 25.29, $SD_{age}$ = 5.39, Range = 18–40; 35 right-handed, 6 left-handed). Participants were recruited via the University of Haifa, the local community, and the Beit Ekstein Center for Adults with Communication Difficulties. All participants completed the Autism Diagnostic Observation Schedule (ADOS) [31] before the experiment, and participants from the autistic group were invited to participate only if the ADOS confirmed their ASD diagnosis. Participants were also matched for IQ using the Test of Nonverbal Intelligence (TONI-4; [32]; ASD (SD, Range): 102.51 (12.65, 84–122)). For additional information on the ASD group see the supporting information. They were compensated for their time at a rate of 80 Israeli shekels per hour. The recruitment period was from December 3, 2020, to March 16, 2022. This study was approved by the local ethics committee of the School of Psychological Sciences at the University of Haifa, with the approval number 358/20. Participants were presented with a detailed consent form outlining the study procedures, and written informed consent was obtained from each participant.

### 2.2 Stimuli and design

**2.2.1 Preferred and estimated distance task (PED [33]).**  Short videos were recorded in high-definition quality at a professional recording studio. These videos featured ten models (five men and five women) instructed to walk toward the camera while maintaining a neutral and consistent facial expression, as well as a uniform pace. The PED task was programmed in E-Prime 3 (version 3.0.3.80).

*2.2.1.1 Preferred interpersonal distance (Fig 1).* Each trial began with a gray 'get ready' screen shown for 1,000 ms, followed by a video clip depicting a stranger either approaching the participants or walking backward away from them. Participants were instructed to stop the stranger at the proximity with which they felt comfortable by clicking the computer mouse to freeze the video. After the video, a 100 ms mask appeared (pixelated mesh of the empty hallway), followed by a gray blank screen shown for 1,000 ms. There were 20 video clips (five men, five women; ten depicting the stranger walking forward, and ten depicting the stranger walking backward). Each stranger covered a total distance of 3.5m at a different pace, yielding clips of 8–12 seconds.

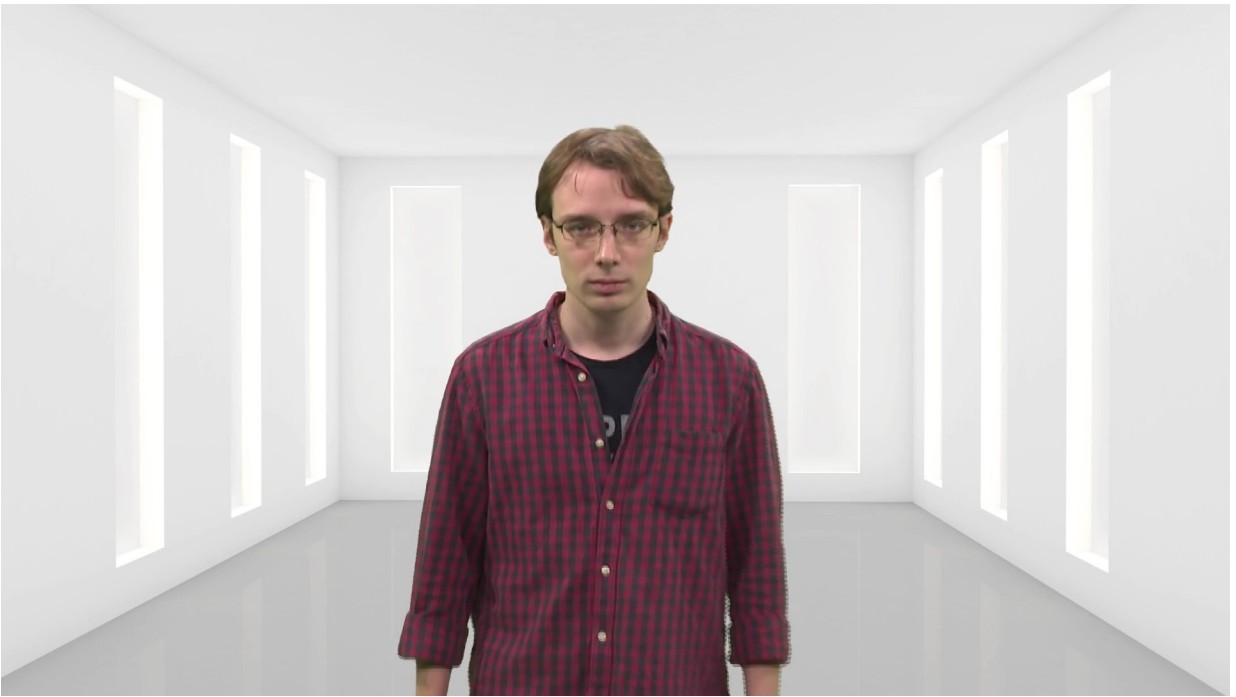

**Fig 1. Example of stimuli from the distance preference task.** Participants viewed a video clip depicting a stranger moving toward them or away from them. In each trial, participants were asked to stop the stranger at the distance with which they felt comfortable. See link for the full videos (https://osf.io/n7yfr/).

The preferred distance score was calculated as follows:

$$Total\ distance\ moved\ in\ video - \left(\frac{Total\ distance\ moved\ in\ video}{Total\ video\ time} * The\ participant\prime s\ RT\right)$$
$$= Preferred\ distance$$

The total distance covered in the video was fixed at 350 cm for all videos. After converting time units (msec) into distance units (cm), the participant's reaction time (RT) was subtracted from this total distance, yielding a preferred distance score in centimeters. The total video time, depicting a man or woman walking toward or away from the viewer, varied from 8 to 12 seconds because each of the models moved at a slightly different pace. The transformation to distance units was based on the assumption that the models maintained a uniform pace throughout the video. Since this transformation is linear (per video), it was similar across participants and did not affect the calculation (i.e., this transformation was applied for convenience). Note that the videos depicting a stranger walking backward were created by playing the walking forward videos in reverse. Hence, for the reversed videos, the preferred distance score was subtracted from 350.

*2.2.1.2 Interpersonal distance estimation bias (Fig 2).* Each trial began with a 'get ready' gray screen shown for 1,000 ms. Afterward, participants were presented with a still photo from one of the video clips depicting a stranger standing at a specific distance from the camera. The image appeared for 1,000 ms, followed by a 250 ms mask and then a video showing the same stranger walking toward or away from the camera from the end of the room. Participants were instructed to stop the video when the stranger reached the exact same distance as in the still photo. The trial concluded with a gray blank screen shown for 1,000 ms.

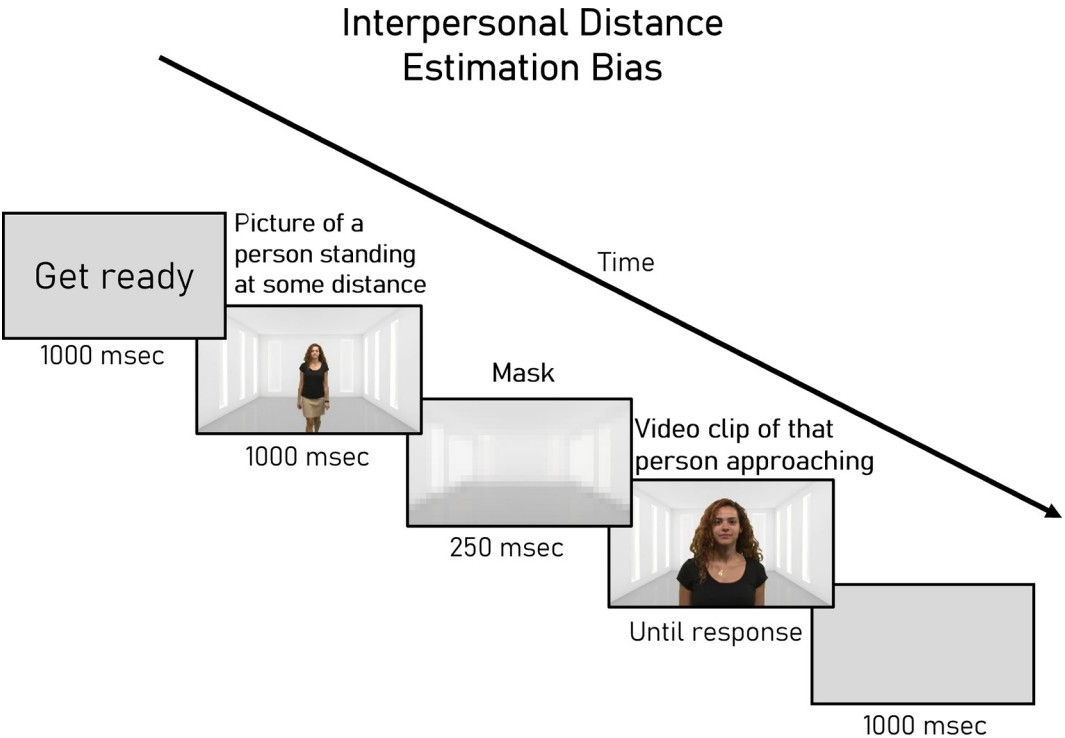

**Fig 2. Distance estimation bias task.** Participants viewed a still photo depicting a stranger standing at a certain distance, followed by a clip depicting the stranger moving toward or away from them. On each trial, participants were asked to stop the stranger at the exact distance as depicted in the still photo.

Four still photos depicting different distances were chosen for each of the ten video clips, yielding 160 images.

The estimated distance bias score was calculated as follows:

$$\left(\frac{Total\ distance\ moved\ in\ video}{Total\ video\ time} * The\ participant's\ RT\right) - \left(\frac{Total\ distance\ moved\ in\ video}{Total\ video\ time} * The\ correct\ answer\ in\ RT\right)$$
$$= Distance\ Estimation\ bias$$

In this formula, the difference between the distance as estimated by the participant was subtracted from the actual distance (in cm). A positive score indicated that the participant estimated the stranger as being closer than they truly were in the still photo, while a negative score indicated that the participant estimated the stranger as being farther away.

*2.2.1.3 Comfort-discomfort scale.* To further examine the validity of the interpersonal distance estimation and preference measures, participants were shown headshots of each of the stranger models and asked to estimate how uncomfortable they felt with their proximity to each model, on a scale ranging from 0 (not at all uncomfortable) to 7 (very uncomfortable). We examined the correlation between the preferred distance and the score on this scale as a manipulation check to validate whether discomfort with proximity was associated with preferred distance.

*2.2.1.4 Motor response.* Participants were shown video clips of strangers approaching or walking away. In these clips, the word STOP would appear at the center of the screen at different time points. Participants were instructed to stop the stranger by pressing the mouse button immediately after seeing the STOP signal. Additionally, some trials involved a vacant room without the approaching stranger, and participants were asked to press the mouse button

upon seeing the word. For each of the ten video clips, four still photos depicting different distances were chosen, resulting in a total of 164 images (four for the vacant room). The motor response score was calculated as the difference in reaction time (RT) between the actual time the word STOP appeared and the participant's response. A negative score indicated a delayed response, while a positive score indicated an impulsive response. This measurement allowed us to account for potential motor differences between the groups and disentangle any confounding factors.

**2.2.2 Questionnaires.**  *2.2.2.1 The autism-spectrum quotient (AQ; [34]).* is a 50-item self-report measure of preferences and tendencies in daily life. Participants rate the extent to which each sentence accurately reflects their preferences and tendencies on a scale ranging from 0 to 3. Scores range from 0 to 50, with higher scores corresponding to a greater number of autistic traits. A clinically significant score on the AQ is 32.

*2.2.2.2 Distance preference due to COVID-19.* To account for potential interference of COVID-related factors, participants were asked the following question: "How much distance do you prefer to maintain from others due to your fear of being infected by the coronavirus?" Participants answered on a three-point scale: "Small Distance" (1), "Medium Distance" (2) and "Large Distance" (3). This response was entered to the analysis as a covariate (please see below).

## 2.3 Procedure

Before starting the experiment, participants positioned themselves in front of the computer screen in a comfortable viewing distance. Participants performed the computerized PED task and then completed the questionnaires. The PED task was divided into three phases administered in sequential order: (1) preferred interpersonal distance; (2) interpersonal distance estimation bias; (3) manipulation check (i.e., comfort\discomfort measure and motor responses), and subsequent questionnaires. After completing the experiment, each participant was debriefed and rewarded.

## 3. Results

### 3.1 Questionnaires

**3.1.1 AQ.**  A significant difference was found between the groups in the scores for the AQ questionnaire ($t$ = 8.807, *Cohen's d* = 2.027, p < .001), with autistic individuals demonstrating higher AQ scores (*M* = 27.02, *SD* = 7.79, *Range* = 13–42) compared to non-autistic individuals (*M* = 14.34, *SD* = 4.64, *Range* = 5–26). In addition to frequentist statistics, the current study utilized Bayesian statistics using analyzed the data with JASP (JASP Team, 2019). In all analyses the default prior was used, with $\alpha$ = 95%. The $BF_{10}$ >100, further confirming that the AQ score was significantly greater in the autistic group. The Error was < 0.001%, indicating great stability. Although this score is rather low, the AQ was still above the cut-off [35].

**3.1.2 COVID-19 item.**  No significant differences emerged between the groups in their COVID-19-related distance preference ratings $t$ = 0.999, *Cohen's d* = 0.223, p = .321, showing that both groups reported similar preferences for distance due to fear of being infected with coronavirus (Autistic: *M* = 1.94, *SD* = 0.60, *Range* = 1–3; non-autistic: *M* = 1.80, *SD* = 0.67, *Range* = 1–3).

## 3.2 Distance preference

To examine the difference between autistics and non-autistics in distance preference, an analysis of covariance (ANCOVA) was conducted, with preferred interpersonal distance as the

dependent variable, group (Autistic/Non-autistic) as a between-subject independent variable, and the item related to COVID-19 as a covariate. No significant difference in distance preference was found between the groups, $F_{(4,76)} = 1.766$, $\eta_p^2 = .085$, $p = .145$ (Fig 3A; Autistic group $M = 194.54$, $SD = 57.91$, $Range = 72.05–353.59$; Non-autistic group $M = 177.17$, $SD = 36.19$, $Range = 103.21–242.30$). However, Levene's test was significant $F_{(1,79)} = 5.041$, $p = .028$, indicating that the variance in the autistic group was greater than the variance in the non-autistic group. Bayesian analysis indicated a $BF_{01} = 2.103$, providing a relatively weak support for the null finding. Furthermore, we also calculated the achieved power of all non-significant comparisons, using G*Power software (version 3.1.9.7; [36]). The power analysis indicated that this comparison was underpowered, with a power of 0.25 (critical $f = 1.274$).

### 3.3 Distance estimation bias

The distance estimation bias analysis was similar to the analyses described above, with interpersonal distance estimation bias as the dependent variable. No significant difference in distance estimation bias was found between the groups, $F_{(4,76)} = 0.114$, $\eta_p^2 = .006$, $p = .977$ (Fig 3B; Autistic group $M = 0.80$, $SD = 16.40$, $Range = -47.10–31.00$; Non-autistic group $M = 0.56$, $SD = 7.98$, $Range = -16.00–23.45$). However, Levene's test was significant $F_{(1,79)} = 5.686$, $p = .020$, again indicating larger variance in the autistic group. Bayesian analysis indicated a $BF_{01} = 4.310$, providing a relatively strong support for the null finding. Post-hoc power analysis indicated a power of 0.97 (critical $f = 0.631$).

### 3.4 Predicting distance preference from distance estimation bias

In order to assess the correlation between distance preference and distance estimation, a hierarchical regression was calculated, with distance preference as the dependent variable and distance estimation bias as the independent variable. The item related to COVID-19 was included as a covariate in a separate step. This analysis was calculated for each group separately, yielding two analyses. Distance estimation bias did not predict distance preferences among individuals in the Non-autistic group, $r = -.150$, $R^2 = .022$, $p = .383$, *Power (1-β) = 0.68*. Bayesian analysis for the non-autistic correlation indicated a $BF_{01} = 2.290$, providing a weak evidence in favor of the $H_0$ hypothesis. In contrast, among autistic individuals, distance estimation bias was positively correlated with the preferred distance, $r = .539$, $R^2 = .290$, $p = .001$, such that estimating the distance as *shorter* was associated with a preference for *closer* proximity (Fig 4). Bayesian analysis for the autistic correlation indicated a $BF_{10} = 34.246$, providing a highly strong evidence in favor of the $H_1$ hypothesis.

Furthermore, the difference between the correlations was examined using the cocor R package [37] for two-tailed tests for independent groups, with an alpha level of 0.05 and a null value of zero. The results indicated that the correlation in the Autistic group differed significantly from the correlation in the non-autistic group ($Z_{Score} = 5.0681$, $p < .001$).

### 3.5 Comfort-discomfort scale

An independent T-test examined group differences in discomfort with proximity to the models in the computerized task. No significant between-group difference in discomfort level was found (t(1,79) = .477, Cohen's d = 0.106, p = .634, Power (1-β) = 0.64), indicating similar discomfort levels in the autistic (M = 3.74, SD = 1.14, Range = 1.00–6.50) and non-autistic groups (M = 3.66, SD = 1.19, Range = 1.00–5.50). Levene's test was not significant (p = .680), indicating equality of variance. Bayesian analysis indicated a $BF_{01} = 2.290$, providing a moderate evidence in favor of the $H_0$ hypothesis.

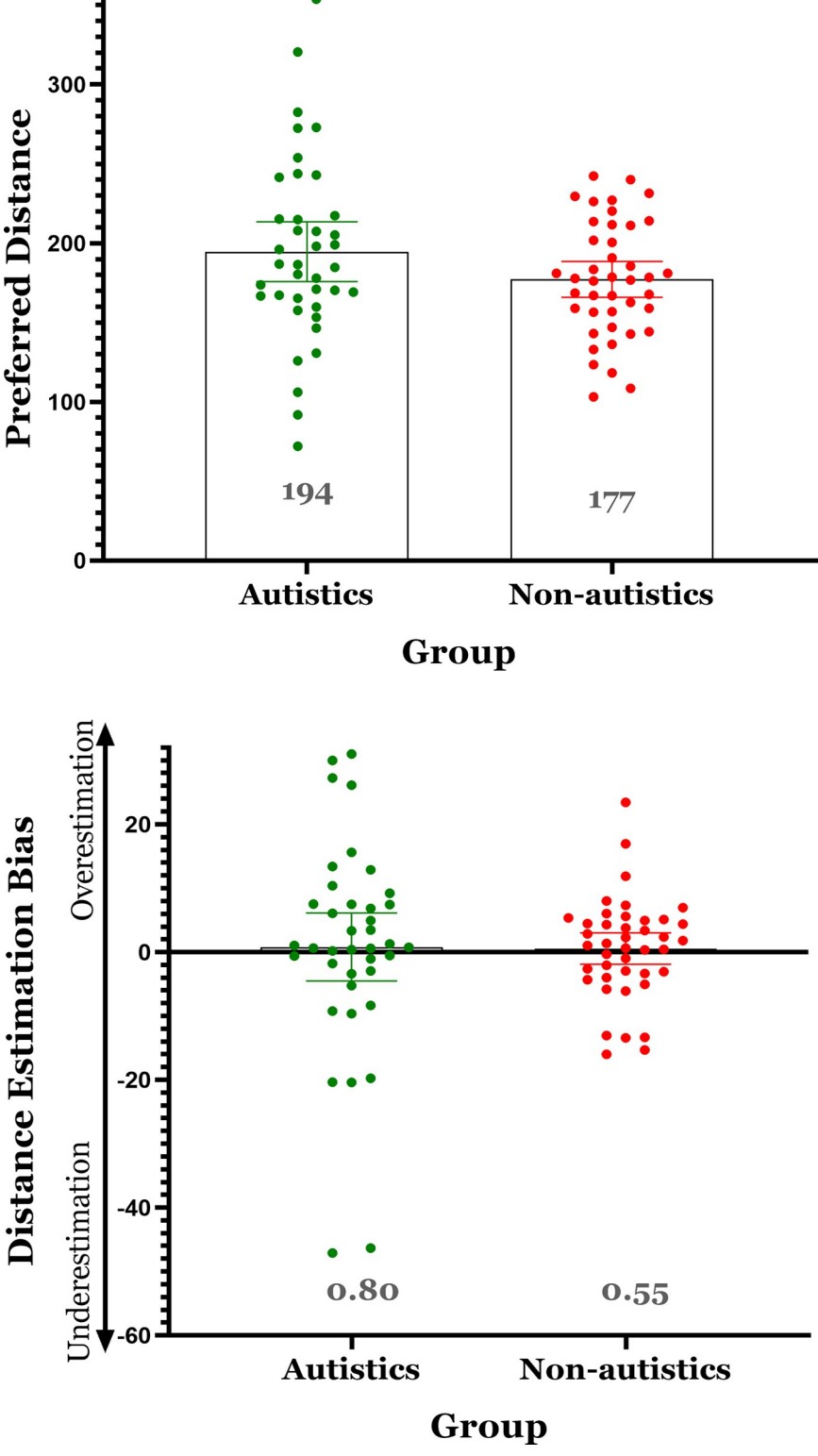

**Fig 3. Means and confidence intervals (95%) as a function of group (autistic versus non-autistic).** (A) Distance Preference, (B) Distance Estimation Bias. Note: The scores of distance preference and estimation bias did not differ between the groups on average, however, in both cases the variance was significantly larger in the autistic group.

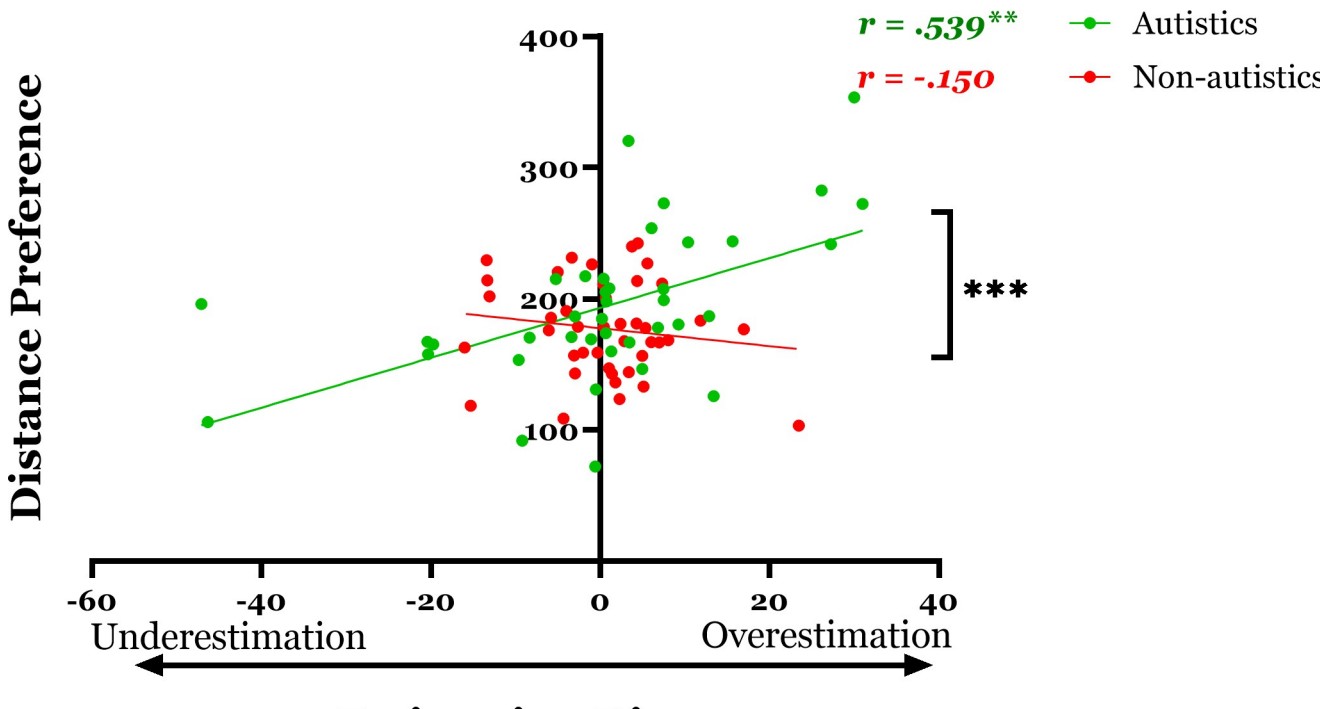

**Fig 4. Preference-estimation association.** Correlation between Distance Estimation Bias (in cm) and Distance Preference (in cm), as a Function of Group. Note: Results indicate that overestimation of distance predicts a preference for greater distance among autistic individuals, and this correlation differs significantly from the non-autistic group participants. (Note: * = p < 0.05, ** = p < 0.01, *** = p < .001).

### 3.6 Motor response

An independent T-test examined between-group differences in the speed of executing the motor response. No significant difference in motor responses was found ($t(1,79) = -0.035$, Cohen's d = -0.008, p = .972, Power $(1-\beta)$ = 0.97). Descriptive statistics also showed only a 1-millisecond difference in reaction time to the STOP signal: Autistic (M = -251.48, SD = 223.27, Range = -1106.21–730.96); Non-autistic (M = -250.24, SD = 63.40, Range = -538.00- -73.42). Levene's test was not significant (p = .052), indicating equality of variance. Bayesian analysis indicated a $BF_{01}$ = 4.322, providing a moderate evidence in favor of the $H_0$ hypothesis.

## 4. Discussion

### 4.1 Results summary

The findings of this study demonstrate that autistic individuals exhibit greater variability in their interpersonal distance preferences and estimations compared to non-autistics. In addition, unlike non-autistic individuals, the preference and the estimation bias were positively linked among autistics. In other words, when autistics underestimate the interpersonal distance, such that the other person is perceived as closer than reality, they tend to prefer a smaller distance. When they overestimate the distance, such that the stranger is perceived as further away, they tend to prefer a larger distance. During the manipulation check, no differences were found between autistics and non-autistics in their level of comfort with the proximity to the strangers or in their motor responses.

## 4.2 Highlighting individual variability: Larger estimation and preference differences in ASD

These results are consistent with previous research indicating greater variability in distance preferences [16–18], highlighting the importance of acknowledging individual differences within the autistic population rather than solely focusing on average differences between individuals with autism and those without. This perspective aligns with recent concerns in the scientific community regarding the oversimplification and stereotyping of individuals on the autism spectrum (e.g., "all autistics are lacking social skills", or "all autistics have superior strengths in math") which ultimately results in overlooking the heterogeneity within this population [37, 38].

The results of the current study demonstrate that variability in distance preference among individuals with autism can be predicted by their ability to estimate distances. This suggests that investigating the relationship between perceptual factors and behavior could be a promising avenue for future research. One unexplored factor is the perception of body size. Evidence suggests that individuals with autism tend to overestimate their own body size [39]. This distorted perception of body size might influence how autistics perceive the spatial relationships between themselves and others. This could lead to a preference for maintaining greater interpersonal distance to accommodate their perceived larger body size. Hence, understanding the factors underlying individual differences in autism would not only help to clarify inconsistencies in the literature, but also to develop individually tailored coping strategies [40].

## 4.3 Exploring the estimation-preference link: Perceptual-motor deficit in ASD?

The current findings suggest that the abnormality in distance regulation does not solely arise from the estimation or preference separately but rather from their interconnected relationship. This interconnection is likely to lead to extreme preferences—either too close or too far—thus violating equilibrium [2]. However, the mechanism that underlies this correlation is unclear.

A possible theoretical framework to explain the relationship between interpersonal distance preference is grounded in a perceptual-motor mechanism. Previous studies have suggested that individuals with autism encounter challenges in integrating perceptual and motor information, termed the perceptual-motor deficit [26, 41, 42]. While autistics may demonstrate accurate distance estimation [21, 22], they might struggle with accurately estimating the distance required to reach out and grasp an object [26, 43]. This difficulty is interpreted as a challenge in translating perceptual information into the motor reaction needed for action. Similarly, autistic individuals also show difficulties in both motor planning [44, 45], and with motor synchrony of interpersonal distance [46]. According to this line of reasoning, difficulty integrating perceptual information may explain the associations between preference and estimation among autistic individuals. However, this mechanism fails to explain the maladaptive-consistent pattern of the effect, such as why autistics approach a closely perceived person and move further away from someone perceived as distant. Therefore, future studies should continue testing the link between sensory processing and behavior. Upon replicating these effects, enriching the theoretical framework to include specific hypotheses regarding the maladaptive correlations between sensory processing and behavior is warranted.

## 4.4 Limitations

**4.4.1 Power analysis.** The current study has several limitations. Firstly, a post-hoc power analysis indicated that the comparison of preferred distances was underpowered. This suggests

that in a larger sample, individuals with autism may indeed exhibit a preference for greater distances and experience greater discomfort in close proximity to strangers compared to non-autistic individuals (with power levels of 0.25 and 0.64, respectively). Therefore, the result indicating indifference in distance preference should be interpreted with caution.

**4.4.2 Tradeoff between virtual and real-life tasks.** A significant tradeoff exists between virtual\computerized tasks and real-life tasks in terms of experimental control versus ecological validity. Virtual studies afford the advantage of conducting multiple trials and utilizing multiple confederates for distance measurement (e.g., 200 trials and 10 confederates in our current study). Conversely, real-life tasks typically involve a small number of trials and usually only one person from whom distance is measured (often the experimenter). Consequently, the preferred interpersonal distance may be influenced by the specific appearance and behavior of the experimenter. Thus, there are distinct advantages to employing virtual tasks, allowing for better control over confounding effects and maintaining consistency across participants.

However, virtual tasks rely on imagination abilities, which may be limited in individuals with ASD [47]. Furthermore, virtual tasks do not encompass all sensory modalities, despite sensory hypersensitivity being a characteristic feature of ASD and closely linked to motor behavior [48]. Additionally, virtual tasks fail to replicate elements of real-life social interactions, such as uncertainty, heightened levels of arousal and stress, and limited opportunities for self-soothing, all of which are significant factors influencing interpersonal distance in ASD [20]. Therefore, future studies should aim to test the replicability and external validity of the findings using real-life paradigms (see comprehensive review [20]) and paradigms that simulate social interactions without the presence of an actual experimenter (see the 'false interview' task [33]).

**4.4.3 Gender distribution.** Another limitation is the gender distribution, with a higher proportion of male participants in the autistic group (76%). Since previous studies indicate gender differences in the presentation of autism [49], we cannot rule out a difference in phenotype in the reported effects. Additionally, it's important to note that besides gender, other environmental, cultural, and sociopsychological factors that were not measured in this study may also influence interpersonal distance (see the review [50]).

**4.4.4 Participant selection and diagnosis confirmation.** While efforts were made to match individuals on nonverbal IQ, it is important to acknowledge that other critical factors such as education level, comorbidities, and medication were not systematically accounted for in the participant matching process. Additionally, while participants provided documentation of their diagnosis from qualified professionals and underwent confirmatory diagnosis using the ADOS evaluation, it is recognized that relying on ADOS scores for inclusion may have limitations. The ADOS alone may not always provide a comprehensive picture of autism diagnoses in adults, particularly in cases involving extensive therapy history or comorbid psychiatric conditions. Furthermore, the relatively low AQ scores observed in our cohort, though still significantly greater in the ASD group, suggest potential complexities in diagnostic reliability of both the ADOS and the AQ in our cohort. While we made an effort to recruit participants based on both the ADOS and their clinical diagnosis and confirm it using the AQ, moving forward, future research should replicate these findings while incorporating comprehensive diagnostic measures and clinical history to ensure accurate participant selection and robust diagnosis confirmation processes, as well as examine potential moderating and confounding factors.

## 5. Conclusions

The current findings suggest that, on average, autistics demonstrate similar interpersonal distance preferences to non-autistics and estimate interpersonal distance with the same level of

accuracy. Interestingly, however, autistics do show greater variability compared to non-autistics, indicating that autistics are characterized by greater individual differences. Additional findings suggest that autistic individuals' preferred interpersonal distance can be predicted by their distance estimation. However, the link between estimation bias and distance preference tends to result in more extreme preferences, potentially violating the equilibrium by being either too close or too far. The combined results highlight the need to examine the relationship between visual processing and social behavior among autistics in order to better understand the mechanisms underlying social deficiencies.

## Supporting information

**S1 File. Additional information on the ASD group.**
(PDF)

## Author Contributions

**Conceptualization:** Nur Givon-Benjio, Bat-Sheva Hadad, Hadas Okon-Singer.

**Data curation:** Nur Givon-Benjio, Tom Marx.

**Formal analysis:** Nur Givon-Benjio.

**Investigation:** Nur Givon-Benjio.

**Methodology:** Nur Givon-Benjio.

**Project administration:** Nur Givon-Benjio.

**Resources:** Marissa Hartston, Idan M. Aderka, Bat-Sheva Hadad, Hadas Okon-Singer.

**Software:** Nur Givon-Benjio.

**Supervision:** Hadas Okon-Singer.

**Visualization:** Nur Givon-Benjio.

**Writing – original draft:** Nur Givon-Benjio.

**Writing – review & editing:** Nur Givon-Benjio, Bat-Sheva Hadad, Hadas Okon-Singer.

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
