## [Decision Letter · Decision Letter 0]

7 Feb 2024

PONE-D-23-39903The Relationship Between Interpersonal Distance Preference and Estimation Accuracy in AutismPLOS ONE

Dear Dr. Givon-Benjio,

Thank you for submitting your manuscript to PLOS ONE. After careful consideration, we feel that it has merit but does not fully meet PLOS ONE’s publication criteria as it currently stands. Therefore, we invite you to submit a revised version of the manuscript that addresses the points raised during the review process.

We look forward to receiving your revised manuscript.

Kind regards,

Claudia Brogna

Academic Editor

PLOS ONE

3. We note that Figures 1 and 2 includes an image of a [patient / participant / in the study].

Reviewers' comments:

Reviewer's Responses to Questions

**Comments to the Author**

1. Is the manuscript technically sound, and do the data support the conclusions?

Reviewer #1: Partly

Reviewer #2: Partly

2. Has the statistical analysis been performed appropriately and rigorously? 

Reviewer #1: Yes

Reviewer #2: Yes

3. Have the authors made all data underlying the findings in their manuscript fully available?

Reviewer #1: Yes

Reviewer #2: Yes

4. Is the manuscript presented in an intelligible fashion and written in standard English?

Reviewer #1: Yes

Reviewer #2: Yes

5. Review Comments to the Author

Reviewer #1: Thank you for the opportunity to review the article titled "The Relationship Between Interpersonal Distance Preference and Estimation Accuracy in Autism" submitted to PLOS One. The study examined an important topic, and addresses a gap in the interpersonal distance literature, by investigating the correlation between distance estimation biases and preferences in autism. I believe interpersonal distance and its regulation is indeed part of the complexities of social communication, it suggests that there is a possibility for autistic individuals to make errors or experience difficulties in this aspect.

The methodology employed in this study is thorough, and the results are presented with clarity.

The manuscript could be considered for publication once the following aspects are addressed, which would significantly enhance the quality of the paper.

1. Introduction:

The literature of interpersonal distance in autism is not sufficiently broad, the literature review may not have been extensive. Although, it is possible that during the lengthy process of writing and review, the authors had not yet been aware of the most recent publications at the time of drafting the introduction, it would have been advisable to conduct a search prior to submission. There are a few new articles from the last year, and it would be interesting to discuss the results in light of the latest theories and results.

- Farkas K, Pesthy O, Guttengéber A, Weigl AS, Veres A, Szekely A, Komoróczy E, Szuromi B, Janacsek K, Réthelyi JM, Németh D. Altered interpersonal distance regulation in autism spectrum disorder. PLoS One. 2023 Mar 31;18(3):e0283761. doi: 10.1371/journal.pone.0283761

- Fusaro M, Fanti V, Chakrabarti B. Greater interpersonal distance in adults with autism. Autism Res. 2023 Oct;16(10):2002-2007. doi: 10.1002/aur.3013

- Farkas K, Pesthy O, Janacsek K, Németh D. Interpersonal Distance Theory of Autism and Its Implication for Cognitive Assessment, Therapy, and Daily Life. Perspect Psychol Sci. 2024 Jan;19(1):126-136. doi: 10.1177/17456916231180593

It might be also interesting to compare the results of interpersonal distance estimation with estimation bias about the self in autism.

- Asada K, Tojo Y, Hakarino K, Saito A, Hasegawa T, Kumagaya S. Brief Report: Body Image in Autism: Evidence from Body Size Estimation. J Autism Dev Disord. 2018 Feb;48(2):611-618. doi: 10.1007/s10803-017-3323-x

2. Methods and study design

a. Participants characteristics: authors described that participants were high functioning and autistics were matched by IQ to neurotypical controls, however education level, estimated IQ, comorbidities were not reported. Comorbidities and medication are especially important.

3. The above aspects are minor issues due to the matching (nonverbal IQ testing), but it is not clear why did the authors decide not to include participants if they did not score positively on the ADOS? ADOS is neither eligible in itself to confirm the diagnosis of autism in adulthood, nor specific. Many of the adult autistics have been in therapy for a long time at the age of 25, and score lower in ADOS, than the threshold, and psychiatric diagnoses (such as psychotic conditions, severe mood disorders, social anxiety etc.) might result in false positives.

The history of early atypical development and clinically confirmed diagnosis could be valid indicators of proper ASD diagnosis. It is especially problematic, as AQ scores are relatively low (which is also possible in itself among autists in care). The authors addressed this issue in section 3.1.1. How can the low AQ score be reconciled with the high ADOS scores that confirm the diagnosis?

4. Data analysis and results: the concept and the statistical analysis is clear and simple

5. Discussion: The cautious interpretation of the results is mostly valid; however, there are two points where I disagree with the authors.

a. In 5.1 pont authors argued that “autists may choose an interpersonal distance based on perceived expectations of others rather than their own personal comfort.“ Most often the exact opposite is the case: autists might not at all consider others’ preference, only choose social interactions (including distance) based on their own personal notion or perspective, ignoring the fact that the other person is already feeling uncomfortable or doesn't even notice the attempt at closeness (being off the radar).

b. The next sentence “when autistic individuals overestimate interpersonal distance, they may anticipate their social partner approaching and, therefore, choose to increase their distance” does not really seem logical to me; in fact, I would assume the opposite correlation. If you overestimate a distance, you would continue to approach until you reach the appropriate and comfortable distance. Overestimating the distance means that you perceive the other person or object to be farther away than they actually are, so you would adjust your movement accordingly to close the gap and reach the desired distance.

6. Limitations: It might be a more pronounced limitation, that the study used a virtual social situation, as the actual presence of the other person has implications on other sensory modalities (olfactory, proprioceptive, acoustic), beyond the challenging situation of meeting a stranger face to face.

7. Finally: There are a few typos in the manuscript; it would be worthwhile to carefully review the text for accuracy (e.g. 4.1. chapter title: Results Summery).

Reviewer #2: The manuscript addresses a highly exciting and important issue: the regulation of interpersonal distance and distance estimation in autism. Due to the limited number of studies in this area, all the collected data are crucial and should be published.

However, there are several points for major revision:

1. Some significant recent results and theories in this area have been omitted from the literature. For example, Farkas et al. (2023) examined interpersonal distance regulation in autism, and their study's ecological validity is much higher than many others, as they investigated distance regulation in real circumstances, not just virtually or using VR. This should be mentioned and interpreted in the introduction and discussion.

2. In 2024, a significant theoretical work was published on distance regulation in autism, which fundamentally defines this area (Farkas et al., 2024). This theory should be integrated into both the introduction and the discussion. I can give a broader perspective to the manuscript.

3. In addition to classical frequentist statistics, the results section should also include the results of Bayesian statistics.

4. It is advisable to rewrite the first paragraph of the discussion to only summarize the results. The interpretation should be proposed in the subsequent paragraphs.

5. One limitation of the study is that the distance regulation was not conducted in real circumstances; instead, the participants looked at a monitor, where a figure approached or moved away, etc. This significantly reduces the ecological validity of the study. The authors mention this in the limitations section, but it should be further elaborated. Sensory hypersensitivity is a very important factor in autism. This is completely overlooked in this type of study design. In the design used by the authors, the autistic participant does not see a real person with all their characteristics, such as random movements, smells, sounds, footsteps, etc. In a more realistic situation, as in Farkas et al. (2023), these are present. It would be worth making a detailed comparison of the different study designs in terms of hypersensitivity and ecological validity.

References:

1. Farkas et al. (2023). Altered interpersonal distance regulation in autism spectrum disorder. Plos one, 18(3), e0283761.

2. Farkas et al. (2024). Interpersonal distance theory of autism and its implication for cognitive assessment, therapy, and daily life. Perspectives on Psychological Science, 19(1), 126-136.

6. PLOS authors have the option to publish the peer review history of their article (what does this mean?). If published, this will include your full peer review and any attached files.

Reviewer #1: No

Reviewer #2: No

---

## [Author Response · Author response to Decision Letter 0]

17 May 2024

Dear Editor and Reviewers,

Thank you for the opportunity to revise and resubmit our manuscript.

We found the reviewers' comments highly constructive and insightful, and we believe that the manuscript is now significantly improved. We appreciate your time and effort in reviewing our paper and hope that you will find your concerns have been adequately addressed.

With the hope that the revised version will meet the high standard of Scientific Reports, we eagerly await your response.

Nur Givon-Benjio (on behalf of all authors)

PONE-D-23-39903

The Relationship Between Interpersonal Distance Preference and Estimation Accuracy in Autism

PLOS ONE

Dear Dr. Givon-Benjio,

Thank you for submitting your manuscript to PLOS ONE. After careful consideration, we feel that it has merit but does not fully meet PLOS ONE’s publication criteria as it currently stands. Therefore, we invite you to submit a revised version of the manuscript that addresses the points raised during the review process.

We look forward to receiving your revised manuscript.

Kind regards,

Claudia Brogna

Academic Editor

PLOS ONE

After reviewing the instructions and observing the option for OSF, I have created a new link containing the minimal dataset necessary: [https://osf.io/n7yfr/]. Please let me know if any further adaptations are required. Thank you.

3. We note that Figures 1 and 2 includes an image of a [patient / participant / in the study].

Please note that the figures do not contain images of participants, as suggested, but rather pictures of models (university students).

Reviewers' comments:

Reviewer #1: Thank you for the opportunity to review the article titled "The Relationship Between Interpersonal Distance Preference and Estimation Accuracy in Autism" submitted to PLOS One. The study examined an important topic, and addresses a gap in the interpersonal distance literature, by investigating the correlation between distance estimation biases and preferences in autism. I believe interpersonal distance and its regulation is indeed part of the complexities of social communication, it suggests that there is a possibility for autistic individuals to make errors or experience difficulties in this aspect.

The methodology employed in this study is thorough, and the results are presented with clarity.

The manuscript could be considered for publication once the following aspects are addressed, which would significantly enhance the quality of the paper.

1. Introduction:

The literature of interpersonal distance in autism is not sufficiently broad, the literature review may not have been extensive. Although, it is possible that during the lengthy process of writing and review, the authors had not yet been aware of the most recent publications at the time of drafting the introduction, it would have been advisable to conduct a search prior to submission. There are a few new articles from the last year, and it would be interesting to discuss the results in light of the latest theories and results.

- Farkas K, Pesthy O, Guttengéber A, Weigl AS, Veres A, Szekely A, Komoróczy E, Szuromi B, Janacsek K, Réthelyi JM, Németh D. Altered interpersonal distance regulation in autism spectrum disorder. PLoS One. 2023 Mar 31;18(3):e0283761. doi: 10.1371/journal.pone.0283761

- Fusaro M, Fanti V, Chakrabarti B. Greater interpersonal distance in adults with autism. Autism Res. 2023 Oct;16(10):2002-2007. doi: 10.1002/aur.3013

- Farkas K, Pesthy O, Janacsek K, Németh D. Interpersonal Distance Theory of Autism and Its Implication for Cognitive Assessment, Therapy, and Daily Life. Perspect Psychol Sci. 2024 Jan;19(1):126-136. doi: 10.1177/17456916231180593

It might be also interesting to compare the results of interpersonal distance estimation with estimation bias about the self in autism.

- Asada K, Tojo Y, Hakarino K, Saito A, Hasegawa T, Kumagaya S. Brief Report: Body Image in Autism: Evidence from Body Size Estimation. J Autism Dev Disord. 2018 Feb;48(2):611-618. doi: 10.1007/s10803-017-3323-x

Thank you for your valuable feedback. We have revised the manuscript to include a broader literature review and discussion of recent publications on interpersonal distance regulation in autism. For example:

-Introduction, page 3 lines 19-27

Recent theoretical work [20] lays out how internal factors (such as sensory hypersensitivity and cognitive processing differences) and external factors (such as social context and cultural norms) can interact to shape the interpersonal distance preferences of individuals with autism. For example, the preferred distance might be determined by autistics' hypersensitivity to sensory modalities, making the smell, sound, and appearance of the social partner much more salient. Farkas et al. [20] also suggest that interpersonal distance regulation plays a role in determining the social and cognitive performance of individuals with autism, further highlighting the importance of studying interpersonal distance in this population. 

Furthermore, we discuss the results of Asada et al., (2018) on body size overestimation in ASD, within our discussion of new avenues to understand individual differences in this population:

-Discussion, page 17 lines 3-13: 

The results of the current study demonstrate that variability in distance preference among individuals with autism can be predicted by their ability to estimate distances. This suggests that investigating the relationship between perceptual factors and behavior could be a promising avenue for future research. One unexplored factor is the perception of body size. Evidence suggests that individuals with autism tend to overestimate their own body size [39]. This distorted perception of body size might influence how autistics perceive the spatial relationships between themselves and others. This could lead to a preference for maintaining greater interpersonal distance to accommodate their perceived larger body size. Hence, understanding the factors underlying individual differences in autism would not only help to clarify inconsistencies in the literature, but also to develop individually tailored coping strategies [40].

Additionally, we have expanded the discussion to address methodological considerations raised by the recent literature, such as the advantages and limitations of virtual and real-life paradigms for measuring interpersonal distance, considering Farkas et al. (2024).

-Discussion (limitations), page 18 lines 18-27, page 19 lines 1-10:

A significant tradeoff exists between virtual\\computerized tasks and real-life tasks in terms of experimental control versus ecological validity. Virtual studies afford the advantage of conducting multiple trials and utilizing multiple confederates for distance measurement (e.g., 200 trials and 10 confederates in our current study). Conversely, real-life tasks typically involve a small number of trials and usually only one person from whom distance is measured (often the experimenter). Consequently, the preferred interpersonal distance may be influenced by the specific appearance and behavior of the experimenter. Thus, there are distinct advantages to employing virtual tasks, allowing for better control over confounding effects and maintaining consistency across participants. 

However, virtual tasks rely on imagination abilities, which may be limited in individuals with ASD [47]. Furthermore, virtual tasks do not encompass all sensory modalities, despite sensory hypersensitivity being a characteristic feature of ASD and closely linked to motor behavior [48]. Additionally, virtual tasks fail to replicate elements of real-life social interactions, such as uncertainty, heightened levels of arousal and stress, and limited opportunities for self-soothing, all of which are significant factors influencing interpersonal distance in ASD [20]. Therefore, future studies should aim to test the replicability and external validity of the findings using real-life paradigms (see comprehensive review [20]) and paradigms that simulate social interactions without the presence of an actual experimenter (see the ‘false interview’ task [33]).

2. Methods and study design

a. Participants characteristics: authors described that participants were high functioning and autistics were matched by IQ to neurotypical controls, however education level, estimated IQ, comorbidities were not reported. Comorbidities and medication are especially important.

Please see our reply to the next comment.

3. The above aspects are minor issues due to the matching (nonverbal IQ testing), but it is not clear why did the authors decide not to include participants if they did not score positively on the ADOS? ADOS is neither eligible in itself to confirm the diagnosis of autism in adulthood, nor specific. Many of the adult autistics have been in therapy for a long time at the age of 25, and score lower in ADOS, than the threshold, and psychiatric diagnoses (such as psychotic conditions, severe mood disorders, social anxiety etc.) might result in false positives.

The history of early atypical development and clinically confirmed diagnosis could be valid indicators of proper ASD diagnosis. It is especially problematic, as AQ scores are relatively low (which is also possible in itself among autists in care). The authors addressed this issue in section 3.1.1. How can the low AQ score be reconciled with the high ADOS scores that confirm the diagnosis?

Thank you for bringing this important issue to our attention. We address this as a limitation in the discussion. Specifically, we acknowledge the issue of not matching participants on other factors besides IQ. Additionally, we discuss the limitation of relying solely on ADOS as a confirmatory diagnosis, as well as the relatively low AQ levels.

-Discussion (limitations), page 19 lines 18-27, page 20 lines 1-6:

4.4.4. Participant Selection and Diagnosis Confirmation

While efforts were made to match individuals on nonverbal IQ, it is important to acknowledge that other critical factors such as education level, comorbidities, and medication were not systematically accounted for in the participant matching process. Additionally, while participants provided documentation of their diagnosis from qualified professionals and underwent confirmatory diagnosis using the ADOS evaluation, it is recognized that relying on ADOS scores for inclusion may have limitations. The ADOS alone may not always provide a comprehensive picture of autism diagnoses in adults, particularly in cases involving extensive therapy history or comorbid psychiatric conditions. Furthermore, the relatively low AQ scores observed in our cohort, though still significantly greater in the ASD group, suggest potential complexities in diagnostic reliability of both the ADOS and the AQ in our cohort. While we made an effort to recruit participants based on both the ADOS and their clinical diagnosis and confirm it using the AQ, moving forward, future research should replicate these findings while incorporating comprehensive diagnostic measures and clinical history to ensure accurate participant selection and robust diagnosis confirmation processes, as well as examine potential moderating and confounding factors. 

In addition, we added additional information on the ASD participants, which is now included in the supplementary materials.

Supplementary:

In the ASD group, IQ was assessed using the Test of Nonverbal Intelligence (TONI-4; Brown et al., 2010), with an average MIQ of 102.51 (SDIQ = 12.65, range: 84-122). Educational attainment within the ASD group included completion of high school by all participants, with 34.6% holding a bachelor's degree and 3.8% holding a master's degree. Regarding comorbidity, 23% reported a diagnosis of Attention Deficit Disorder (ADD) or Attention Deficit Hyperactivity Disorder (ADHD), 11% reported Obsessive-Compulsive Disorder (OCD), and 3% reported borderline personality disorder. Additionally, 48% reported no comorbidity, while the rest preferred not to disclose this information. Concerning medications, 50% reported receiving medication for ADHD, a mood disorder, or both, while 38% were 

---

## [Decision Letter · Decision Letter 1]

20 Jun 2024

The Relationship Between Interpersonal Distance Preference and Estimation Accuracy in Autism

PONE-D-23-39903R1

Dear Dr. Nur Givon-Benjio,

We’re pleased to inform you that your manuscript has been judged scientifically suitable for publication and will be formally accepted for publication once it meets all outstanding technical requirements.

Kind regards,

Claudia Brogna

Academic Editor

PLOS ONE

Reviewers' comments:

Reviewer's Responses to Questions

**Comments to the Author**

1. If the authors have adequately addressed your comments raised in a previous round of review and you feel that this manuscript is now acceptable for publication, you may indicate that here to bypass the “Comments to the Author” section, enter your conflict of interest statement in the “Confidential to Editor” section, and submit your "Accept" recommendation.

Reviewer #1: All comments have been addressed

Reviewer #2: All comments have been addressed

2. Is the manuscript technically sound, and do the data support the conclusions?

Reviewer #1: Yes

Reviewer #2: Yes

3. Has the statistical analysis been performed appropriately and rigorously? 

Reviewer #1: Yes

Reviewer #2: Yes

4. Have the authors made all data underlying the findings in their manuscript fully available?

Reviewer #1: Yes

Reviewer #2: (No Response)

5. Is the manuscript presented in an intelligible fashion and written in standard English?

Reviewer #1: Yes

Reviewer #2: Yes

6. Review Comments to the Author

Reviewer #1: The authors have addressed the reviewers' comments, the quality of the revised article has improved and now suitable for publication.

Reviewer #2: (No Response)

7. PLOS authors have the option to publish the peer review history of their article (what does this mean?). If published, this will include your full peer review and any attached files.

Reviewer #1: No

Reviewer #2: No

---

## [Editor Report · Acceptance letter]

24 Jun 2024

PONE-D-23-39903R1 

PLOS ONE

Dear Dr. Givon-Benjio, 

I'm pleased to inform you that your manuscript has been deemed suitable for publication in PLOS ONE. Congratulations! Your manuscript is now being handed over to our production team.

Kind regards, 

on behalf of

Dr. Claudia Brogna 

Academic Editor

PLOS ONE